# Hydrogel Based on Tricarboxi-Cellulose and Poly(Vinyl Alcohol) Used as Biosorbent for Cobalt Ions Retention

**DOI:** 10.3390/polym13091444

**Published:** 2021-04-29

**Authors:** Iulia Nica, Carmen Zaharia, Daniela Suteu

**Affiliations:** 1Department of Organic, Biochemical and Food Engineering, ‘Cristofor Simionescu’ Faculty of Chemical Engineering and Environment Protection, “Gheorghe Asachi” Technical University of Iasi, 73 D. Mangeron Blvd., 700050 Iasi, Romania; nebunu.iulia@gmail.com (I.N.); danasuteu67@yahoo.com (D.S.); 2Department of Environmental Engineering and Management, ‘Cristofor Simionescu’ Faculty of Chemical Engineering and Environment Protection, “Gheorghe Asachi” Technical University of Iasi, 73 D. Mangeron Blvd., 700050 Iasi, Romania

**Keywords:** biosorbent, Co(II) ion, hybrid cellulose hydrogel, kinetic modeling

## Abstract

A biomaterial based on poly(vinyl alcohol) reticulated with tricarboxi-cellulose obtained by TEMPO oxidation (OxC25) was used as a new biosorbent for Co(II) ions retention from aqueous solutions. The biosorption process of Co(II) ions was studied while mainly considering the operational factors that can influence it (i.e., biosorbent concentration, pH of the aqueous media, temperature and contact time of the phases). The maximum adsorption capacity was 181.82 mg/g, with the biosorption well fitted by the Langmuir model. The kinetic modeling of the biosorption process was based on certain models: Lagergreen (pseudo first order model), Ho (pseudo second order model), Elovich (heterogeneous biosorbent model), Webber–Morris (intraparticle diffusion model) and McKay (film diffusion model). The corresponding kinetic model suggests that this biosorption process followed a pseudo-second order kinetic model and was developed in two controlled steps beginning with film diffusion and followed by intraparticles diffusion.

## 1. Introduction

Clean water is strongly needed for humanity, animals and plants on a planet with more and more diminishing natural water resources. Society has learned that for prevention of disease and reduction of environmental pollution levels, all wastewater produced must be treated and even reused/recycled to save fresh water supplies [1,2,3].

Water quality management and its subareas of water and wastewater treatment are a key issue in environmental policy; therefore, the performance of different water/wastewater treatments must be permanently controlled and improved [4,5,6].

Metals are important nutrients required in substantial amounts by plants, animals and microorganisms, and are included in the significant category of environmentally important cations, i.e., Na^+^, K^+^, Mg^2+^, Ca^2+^ and Al^3+^. Other metals such as Cu^2+^, Zn^2+^ and even Co^2+^ are also nutrients, but the amounts required by organisms are very small, and if are present in excessive amounts can be toxic (even when forming chemically stable complexes). For example, for Co^2+^ species, the chronic intoxication of human organisms is due to ingestion caused by the consumption of contaminated water and/or permanent dermal contact with it (inducing intense itching of the skin), and located (30%) in some human body organs with lent in-time elimination, and expressed in the form of pronounced hypotension, stomach pain, digestive bleeding, allergic eczema and mucosal irritation [7,8].

Some metals present in water/wastewater are naturally retained by physical entrapment, complexation with different free ligands (e.g., chloride, carbonate, citrate, fulvic acid, etc.) or incorporated into the cellular structures, and further artificially removed by the use of chemicals such as coagulants-flocculants (alum or ferric chloride) followed by filtration through a microscreen or sand bed, precipitation after chemicals addition, adsorption (onto granular activated charcoal—GAC or other adsorbent materials), ion exchange, reverse osmosis, etc.

Metal species stability in water/wastewater depends on a number of factors such as: (i) the nature of metal ion species (e.g., Co(II) ions forming aqua complexes being protonated or partially deprotonated in the aqueous media as Co(H_2_O)_6_^2+^, with a relatively high value of instability constant—pK_a1_ = 9.6, or Co(H_2_O)_5_(OH)^(+)^, or complexes with free ligands); (ii) ambient solution pH and redox status (oxidizing and reducing conditions); (iii) ionic strength (dependent of mineralization degree, total salts content), and (iv) availability of functional groups (i.e., a heterogeneous collection of functional groups of the present free organic or inorganic ligands in water) [7]. Typical values for the heavy metal complexation capacity are 1–2 μmol/L in rivers, 2–5 μmol/L in lakes, 5–15 μmol/L in ponds or >15 μmol/L in swamps [3]. As an example, the background concentration of cobalt-based species in fresh water (2nd quality class) is about 10 μg/L in the form of Co^2+^ ions and/or CoCO_3_, with a small portion as Co_3_O_4_ species or of 3−5 μg/L in open ocean sea water, and the maximum admissible concentration of Co^2+^ ions in treated wastewater (final effluent) discharge in fresh water (natural aquatic receptor) is 1 mg/L [9].

Adsorption continues to be among the methods often applied to retain certain organic or inorganic pollutants, mainly due to advantages that are difficult to neglect: equipment for applications that are easy to design and operate under advantageous conditions, the possibility of using a wide range of cheap adsorbent materials (‘low cost’ adsorbents) and applications depending on the type of pollutant and the required process conditions. Due to their adsorbent properties, biomaterials can be used as (bio)adsorbents in adsorption processes applied for the removal of pollutants from water/wastewater (e.g., dyes, phenol, pesticides, toxic metal ions, drug residues) [10,11,12,13,14].

Polysaccharides have adsorbent properties, which are exploitable in concentration-separation-purification processes as (bio)adsorbents with high specificity [15,16,17,18,19,20,21,22,23].

A new type of adsorptive biomaterials is represented by the hydrogels—a polymeric structure which contains hydrophilic and hydrophobic fractions in a defined proportion, convenient swelling capacity in the water, hydrophilicity, biocompatibility and a lack of toxicity which encourage their use in various fields, from sanitary, agricultural, medical and pharmaceutical applications to those in the area of environmental protection. Thus, there has been an increasing interest in the use of hydrogels for wastewater treatment due to their high adsorption capacities, regeneration facilities and possibilities to be reused in continuous technological processes [22,24,25,26,27,28]. These biomaterials have also a number of disadvantages, mainly related to low mechanics, stability and strength. In order to remedy and thus improve the (bio)adsorptive properties, a number of technological processes have been studied and developed, such as crosslinking methods with auxiliary agents (i.e., crosslinking agents) [24,25,29,30].

A number of studies have focused on poly (vinyl alcohol) (PVA) hydrogels, a biodegradable polymer used in recent years to make hybrid biodegradable materials. Although it has low mechanical strength, a number of components capable of improving its properties have been studied. Research has focused on certain natural polymers (cellulose and pullulan) in different concentrations, in the form of oxidized C6 derivatives, as key multifunctional components that could act as crosslinking agents, but also as agents for increasing the hardening and stiffness properties of hybrid PVA hydrogels [23,26,27,31,32]. Moreover, these types of (bio)adsorbents obtained from natural resources can be successfully used for retaining of some heavy metal species from different aqueous media.

The aim of this paper is to evaluate the biosorptive properties towards Co^2+^ ions of new (bio)adsorbent represented by a hydrogel (OxC25) based on a matrix of poly(vinyl alcohol) (PVA) with reticulated tri-carboxy cellulose. The biosorption studies were carried out in the following directions: (i) the determination of operational parameters influencing the biosorption process and their adequate values; (ii) the determination of most important kinetic biosorption parameters and the corresponding kinetic biosorption model; and (iii) the design of a possible biosorption mechanism (especially biosorption controlled steps).

## 2. Experimental

### 2.1. Materials

*Biosorbent*: we used a hybrid hydrogel with two basic components: poly(vinyl alcohol) (PVA)and 2,3,6 tricarboxycellulose, prepared by oxidation of the wet cellulose pulp in the presence of the 2,2,6,6-tetramethyl-1-piperidinyloxy (TEMPO)/NaBr/NaOCl system. This biosorbent was obtained from a collaboration with the “P.Poni” Institute of Macromolecular Chemistry, Iasi, Romania. The main characteristics of the two basic components are: *PVA* with MW = 8.9 × 10^4^–9.8 × 10^4^ g/mol, 99% hydrolyzed, and *cellulose* (microcrystalline, Avicel^®^ PH 101), with a degree of polymerization (DP) of 135 (from Sigma-Aldrich Co).

TEMPO, sodium bromide and 9% (wt) sodium hypochlorite were used as commercially available pure grade chemical reagents (Sigma Aldrich Co., St. Louis, MO, USA).

*Metal ionic species*: Co^2+^ ions were selected as polluting inorganic species, presented as a CoSO_4_·7H_2_O solution and tested in the concentration range of 36.62–207.52 mg of Co^2+^/L of solution. The concentration of stock solution was 10^−2^ M Co^2+^.

### 2.2. Methods

#### 2.2.1. Preparation and Physico-Chemical Characterization of Hybrid Hydrogel

The preparation and the physical-chemical characterization of the hydrogel used as a (bio)adsorbent in this study was performed according to the method and protocol described in previous papers [32,33,34], meaning in two stages: (i) the synthesis of 2,3,6-tricarboxycellulose by oxidation in the presence of the TEMPO/NaBr/NaOCl system; and (ii) the proposed stage of hydrogel synthesis, by mixing the homogeneous solution of PVA with the 2,3,6-tricarboxycellulose solution in different ratios (Figure 1). The method continued with three freezing-thawing cycles applied to the composites of PVA-V mixture and pure aqueous PVA solution in order to obtain the physical networks. Notably, after each freezing cycle of the samples, it was provided the operational conditions for a slow thawing by keeping them at 4 °C, which ensures the formation of a porous structure [23,32,33].

For the characterization of the prepared biomaterial, in order to highlight the internal structure and the functional groups responsible for the adsorbent properties, the study of the degree of swelling was performed by using physico-chemical instrumental analysis methods, i.e., Scanning Electron Microscopy (SEM) and Fourier–Transform Infrared Spectroscopy (FTIR), and are described in other authors’ reports [32,33,34]. It must be mentioned that at a very low pH (1.5–2.0), the prepared hybrid hydrogel stability was good, and the swelling degree had the lowest value related to the maximal swelling degree shown at a pH of around 6.5–7.5 [35].

#### 2.2.2. Batch Biosorption Methodology

For study of the biosorption equilibrium and its kinetics, the following procedure was followed: established volumes (25 mL) of Co^2+^ solutions, with different initial concentrations (36.62–207.52 mg/L) and pH values of 1.5 adjusted with the help of H_2_SO_4_ 1N solutions, were contacted with a known amount of biosorbent (0.4 g/L) for 24 h, or a certain time period (*t*) at three temperatures of 5 °C, 20 °C and 35 °C, respectively. The temperature was kept constant using a Poleko SLW 53 thermostatic bath. After 24 h, when it was assumed that equilibrium was reached, or after a certain contact time period (*t*), the residual metal ion (Co^2+^) concentration in aqueous medium was determined using an UV-VIS Digital Spectrophotometer, model S 104D/WPA by reading the aqueous solution absorbance at wavelength λ = 450 nm, after sample treatment with rubeanic acid of 0.05% in borate buffer (pH = 2.5) [36]. The quantitative biosorption characteristics were evaluated by means of the amount of biosorbed Co^2+^ ions and, consequently, biosorption capacities of the hydrogel (Equation (1)):
(1)$$q_{t}=\frac{C_{0}-C_{t}}{G}\cdot V$$
where *C*_0_, *C_t_* are the initial values, after *t* biosorption time of Co^2+^ ion concentration in aqueous medium (mg/L), *G* is the amount of biosorbent (g) and *V* is the volume of the aqueous sample (L).

### 2.3. Fourier Transform Infrared (FTIR) Spectroscopy

The FTIR spectra of samples were recorded in the Attenuated Total Reflection (ATR) configuration on a Golden Gate (diamond crystal, Specac Ltd.) accessory using a Vertex 70 spectrometer (Bruker, Germany) [32]. The spectra were collected as 128 scans co-added at a resolution of 2 cm^−1^, in the medium IR range (4000–600 cm^−1^). Data processing and manipulation was performed with OPUS 6.5 software [32].

### 2.4. Kinetic Modeling of the Biosorption Process

The kinetic data that were experimentally obtained were processed using certain representative kinetic models, as presented below [37,38,39]:
***Pseudo-first order*** (Lagergreen model) is well fitted for modeling of kinetic data if the plot *lg* (*q_e_* −*q_t_*) versus *t* gives straight lines which pass through the origin of axes, and the *q_e exp_* values agree with the calculated *q_e,exp_*. Its general equation is:
(2)$$\frac{dq_{t}}{dt}=k_{1}\cdot \left(q_{e}-q_{t}\right)$$And the linearized form is:
(3)$$\mathrm{lg}\left(q_{e}-q_{t}\right)=lg\;\;q_{e}-\frac{k_{1}}{2.303}\;t$$
where: *k*_1_- the rate constant of pseudo-first order model, 1/min; *q_t_* and *q_e_*—the amount of dye adsorbed at *t* time and at equilibrium (24 h), mg/g, respectively.***Pseudo-second order model*** (Ho model) suggests that the biosorption processes can be controlled by chemical biosorption, or chemo-sorption involving valence forces through sharing or exchanging of electrons between the two phases involved, if it can observe a linearity of plots for *t*/*q_t_* versus *t*, and the values of *q_t_* calculated are much closer to the experimental values. The general equation and its linearized form are:
(4)$$\frac{dq_{t}}{dt}=k_{2}\cdot {\left(q_{e}-q_{t}\right)}^{2}$$
(5)$$\frac{t}{q_{e}}=\frac{1}{k_{2}\cdot q_{e}^{2}}+\frac{1}{q_{e}}\;t$$
where: *k*_2_—the rate constant of pseudo-second order model, g/(mg min); *k*_2_*q_e_*^2^ = *h*—the initial adsorption rate, mg/(g min).***Elovich model*** is applied in an adsorption system in which the adsorbent surface is heterogeneous. The studied processes can be described as a chemical adsorption if the representation *q_t_* versus *ln t* is linear and intersects with the origin of the axes. The characteristic equation is:(6)$$q_{t}=\frac{\ln\left(\propto \cdot \beta \right)}{\beta }+\frac{1}{\beta }\ln t$$
where: *α*—a constant about the initial adsorption rate, mg/(g min); *β*—a constant that refers to the extent of surface coverage and the activation energy for chemo-sorption, g/mg.***Intraparticle diffusion model*** (Webber–Morris model) supposes diffusion-controlled kinetics; pore diffusion is the rate-limiting step if the plot *q_t_* vs. *t*^1/2^ is linear and passes through the origin:
(7)$$q_{t}=k_{d}\cdot t^{1/2}+c$$
where *k_d_* is the rate constant for intraparticle diffusion, mg/(gmin^1/2^); *c* is the intercept to the Y axis.***Film diffusion model*** (McKay model): the film diffusion is involved in the adsorption process if the plot *ln* (*1-F*) vs. *t* is linear, and it is the rate-limiting step if the line passes through theorigin
(8)$$\ln\left(1-F\right)=k_{f}\;t$$
where *k_f_* is the rate constant for film diffusion, 1/min, and *F* is the fractional attainment.


## 3. Results and Discussion

### 3.1. Biosorbent Characterization: Initial and After Co^2+^ Biosorption

The FTIR spectrum was made for the hydrogel sample after the retention of Co^2+^ ions and was compared with the FTIR spectrum for the hydrogel sample before biosorption (Figure 2). The peaks of interest were spectrally resolved by a curve fitting procedure using the Voigt function. The exact position and estimated number of sub-bands were determined using the second derived spectrum, using the Savitzky-Golay algorithm with a 30-point uniformity factor.

The FTIR spectrum of the initial (OxC25) sample, Figure 2, shows the specific absorptions of the two components of hydrogel: polyvinyl alcohol (PVA) and 6-carboxy cellulose (OxC), with maximum absorption at 3327 cm^−1^ (OH groups), 2938 cm^−1^, 2914 cm^−1^ (vibrations of CH and CH_2_ groups) and 1607 cm^−1^ (vibrations of carbonyl bonds in carboxylic groups coming from oxidized cellulose), the combined deformation band δ(CH_2_/CH + OH) with peaks at around 1421 cm^−1^, 1375 cm^−1^ and 1335 cm^−1^ and the fused ν(COH) + ν(COC) + ν(CCH) bands in the region of 1159–1060 cm^−1^. After the (bio)sorption process, the FTIR spectrum undergoes transformations caused by the process of adsorption of Co^2+^ ions on the hydrogel, as follows: the band corresponding to the OH groups changes its allure (3339 cm^−1^), being diminished in intensity, as well as those corresponding to the CH and CH_2_ groups(2895 cm^−1^), or carbonyl bonds in carboxylic groups (1644 cm^−1^), or δ(CH_2_/CH + OH) deformation band (peaks at 1427 cm^−1^, 1368 cm^−1^, 1337 cm^−1^ and 1314 cm^−1^), or the fused ν(COH) + ν(COC) + ν(CCH) bands (1203 cm^−1^, 1159 cm^−1^, 1107 cm^−1^, 1053 cm^−1^). It is also observed the appearance of a high intensity band at 1029 cm^−1^ attributed to the Co^2+^ binding of the PAV-6-carboxy cellulose polymer network, as well as the 665 cm^−1^ band (-Co-O-).

### 3.2. Biosorption Studies

#### 3.2.1. Evaluation of Some Operational Parameters Influencing the Biosorption Process

An in-depth and exhaustive study of the biosorption process, which includes the study of equilibrium in static and dynamic conditions, of the kinetics and thermodynamics of the studied process and also the optimization of the process in static and dynamic conditions, is possible after a series of operational parameters studies that influence the process in both variants (static and dynamic ones). Thus, establishing the way/mode in which these operational parameters influence the development of the biosorption process in association with the structural characteristics of the biosorbent and the adsorbed chemical speciesis the first step in approaching the biosorption process.

The process of Co^2+^ ions biosorption can be influenced by a number of physical operational parameters that can interfere with the establishment of specific ionic functional groups in the structure of the (bio)adsorbent, or the shape of the metal ion (i.e., Co(II) species), the retention capacity rate and the biosorption process nature.

#### 3.2.2. Influence of Solution pH

This operational parameter influences both the behavior of the solid organic material used as a biosorbent (modification of the dissociation degree of functional groups from the structure of the biosorbent which determines a certain charge of their surface) and of inorganic species, i.e., Co^2+^ ions, which must be retained from aqueous media. As a function of the pH value and the charge of the biosorbent surface, it would be possible to accordingly retain the anionic or cationic species from the aqueous solution.

As presented in Figure 3, the amount (q, mg/g) and the percent (R%) of Co^2+^ ions retained, as a function dependent of the initial solution pH, suggested that the Co^2+^ ions could be better biosorbed from acidic media with a pH of around 1.5 (or in a range of 1–2). This finding can be explained taking into account the structure of the hydrogel and the following aspects: at a very acidic pH, the active carboxyl groups (COOH) and their carbonyl groups (CO) of studied biosorbent can attract the Co^2+^ ions on its charged surface and form covalent bonds by metal coordination with chelates formation, -O-Co-O-) thus retaining almost all metal species from the aqueous solution.

Also, this behavior may be correlated with the variation of the hydrogel surface charge in the function of the solution pH. The value of pH_PZC_ (pH of zero charge) for OxC25 hydrogel determined by method proposed by Nouri and Haghseresht [40], was found to be 5.4. At values of pH <pH_PZC_, the hydrogel surface was positively charged due to the increased H^+^ ions concentration [26] (carboxylic groups are predominantly protonated) and susceptible to reacting with anionic species by electrostatic interactions and hydrogen bonding, or discretely associating the metal ions contaminants by ion-exchanges (cations exchange) and chelation between Co^2+^ ions and ionized/non-ionized carboxylic groups within the OxC25 hydrogel at acid pH (1.5–5.4) (e.g., Co^2+^ ions replace the H^+^ from –COOH groups by cation exchange and chelation/coordination with Co-chelates formation). Therefore, the Co^2+^ ions were adsorbed on the hydrogel surface (firstly outside due to mass transport phenomena) and further deeply trapped in the hydrogel structure by cation-exchange and chelation (inside of OxC25 hydrogel chains). At pH >pH_PZC_ values, the sorbent surface is negatively charged due to dissociation of the –COOH functional groups and is capable of anion-exchanges and/or electrostatic interactions with cationic species. In the case of Co^2+^ ions retaining from aqueous solution on OxC25 hydrogel at alkaline pH, the experimental data indicated low removal, which can be explained by not having predominant electrostatic interactions due to anion-exchanges and also by hydrophobic interactions. Moreover, the experimental data indicated very good adsorption properties of OxC25 hydrogel for Co^2+^ ions at very acidic pH values due to the synergetic effects of physical adsorption and chemo-sorption by cation exchanges and coordination/chelation.

#### 3.2.3. Influence of the Biosorbent Concentration

To establish the optimal concentration of biosorbents, varying amounts of tested hydrogel were contacted with 25 mL of Co(II) solutions with definite concentrations. The experimental study was performed under the following operational conditions: a pH value established in previous experimental studies, a temperature of about 20 °C and a phases contact time of 24 h.

The obtained results are presented in Figure 4.

From Figure 4, it can be observed that an increase of the hydrogel concentration from 0.4 to 2.0 g/L induces an increase of the percentage of Co(II) ion removal from 65.27% to 75.88% at the biosorbent concentration of 1 g/L followed by a decrease of Co^2+^ ion removal to 59.28% at 2 g/L, but also a decrease of the amount of reactive dye retained per unit weight of adsorbent from 199.177 mg/g to 36.182 mg/g. This behavior can be explained by the reduction of diffusion inside the solid biosorbent mass due to the some cation steric effects and also the better retention of Co(II) species only at the external surface of biosorbent towards the inside solid structureby cations exchange and chelation. The biosorbent concentration in order to obtain a biosorption capacity as high as possible is 0.4 g/L, respectively, 0.01 g OxC25 hydrogel per 25 mL sample with Co^2+^ ions.

#### 3.2.4. Influence of Initial Co(II) Ions Concentration and Temperature

The biosorption capacity of the studied hydrogel, OxC25, for the selected cation (Co^2+^) was studied in aqueous solutions with the initial pH of 1.5, using aqueous Co(II) solutions with different initial concentrations, from 36.62 to 207.52 mg/L and at three different temperatures (5 °C, 20 °C and 35 °C). The graphical representation of the equilibrium concentration as a function of temperature and the (bio)sorption isotherms, respectively, are shown in Figure 5.

From Figure 5, it can be observed that the amount of retained Co(II) cations in the temperature interval of 5–35 °C increases with the increase of the initial Co(II) concentration which is in contact with the biosorbent. Also, it can be observed that the retained Co^2+^ amount depends on the temperature at which the biosorption is performed. The process is influenced by rising temperature, which suggests that it could be an endothermic process in this considered temperature range.

Also, from Figure 5, it is evident that the biosorption of the Co^2+^ ions onto this hydrogel corresponds to the type “L3” in Gills’s classification system [41]. This behavior shows an increase of the biosorption efficiency and adsorption capacity with the increase of the temperature, suggesting that the process could be endothermic. The maximum biosorption capacity, for a 20 °C tested temperature, was found to be 181.82 mg/g (from Langmuir (bio)adsorption isotherms—data to be published).

#### 3.2.5. Influence of Phases Contact Time

The physical operational parameters of the biosorption process include the contact time between the involved phases: aqueous Co^2+^ ions and solid (bio)adsorbent. Its importance derives from the fact that the phases contact time is useful in the kinetic modeling of the process in order to understand the mechanism and the process type, but it also intervenes in the optimization designof this (bio)sorption process.

The influence of phases contact time on the Co^2+^ ions retaining onto OxC25 hydrogel is presented in Figure 6.

The graphical representation from Figure 6 shows that the values of the fractional attainment of equilibrium ($F=\frac{q_{t}}{q}$) increase rapidly with the phases contact time during the first 250 min and, after that, the rate of Co(II) ions biosorption became slower; thus, the time period required for maximum removal and/or equilibrium establishment was found to last up to 12 h. Accordingly, the (bio)sorption capacity values increased rapidly during the first 120 min and after 500–700 min was almost stabilized, indicating that biosorption equilibrium is almost ready to be established.

### 3.3. Kinetic Modeling

Scientific studies consider that an adsorption process at the solid-water interface can be done by the following three important steps, of which the one that proceeds at the lowest speed will govern the adsorption kinetics [37,42]:
(1)the diffusion of the adsorbate molecules from aqueous medium to the biosorbent surface through the boundary layer (film diffusion),(2)the diffusion of adsorbate molecules from the surface into the pores of the solid particles (pore diffusion, or intraparticle diffusion),(3)the interaction of adsorbate with the active sites on the surface of the biosorbent.

In biosorption processes, the most important step is the diffusion in the liquid film surrounding the biosorbent particles (external diffusion) and the diffusion in the biosorbent particle (internal diffusion), and therefore to investigate the type of biosorption mechanism involved in the retaining of Co^2+^ ions on OxC25 hydrogel-based biosorbents (intraparticle diffusion/film diffusion or chemical interaction) and also to establish the decisive stage of the biosorption process (i.e., the stage with the lowest speed/rate).

The kinetic parameters for each model applied (Equations (2)–(8)), taking into account the graphical representations from the Figure 7, are presented in Table 1. Using the values of the linear regression correlation coefficient, R^2^, we estimated the fit of each model to the experimental data.

The data presented in Table 1 in association with the graphical representations from Figure 7 underline some aspects of the studied kinetics of biosorption process:
It can observed that the biosorption process took place in the most adequate/efficient conditions at a moderate concentration of Co(II) ions.Because the plot *ln* (*q_e_*-*q_t_*) versus *t* for both initial Co(II) concentrations does not give a straight line (Figure 6a,b) and the R^2^ is lower than 0.95, the viable suggestion is that the pseudo-first order model is not well fitted for the modeling of kinetic data.Due to the linearity of plots *t*/*q_t_* versus *t* for both initial Co^2+^ dications concentration and the highest values of the correlation coefficient, *R^2^*, we could suggest that the biosorption kinetics of Co^2+^ ions onto OxC25 hydrogel follow a pseudo-second order kinetic model, and the biosorption process can be controlled by chemical adsorption, or chemo-sorption involving valence forces through sharing or exchange of cations and electrons between the two phases involved and also coordination/chelation.R^2^ values less than 0.95 for this studied adsorption system (Table 1) and the fact that the linear representations *q_t_* versus *ln t*, for both initial concentrations of Co^2+^ (Figure 7d), do not intersect with the origin of the axes, suggest that the data do not fit well with the Elovich equation, emphasizing that chemo-adsorption may not be the only step that controls the studied (bio)sorption process. The hypothesis remains that the diffusion process is the stage that could control the biosorption.Taking into account these three conclusions, it could be assumed that the biosorption rate can be governed by either liquid phase mass transport, or by intraparticle mass transport. In order to obtain more accurate information about the diffusion mechanism, the kinetic data were analyzed by the *intra-particle diffusion model* (by the Webber–Morris model) (Figure 7c) and *film diffusion model* (by McKay model) (Figure 7e) [37,43]. In Figure 7c the experimental points associated with the Webber–Morris model are positioned on a straight line, then intra-particle diffusion occurs, but since neither passes through the origins, it is suggested that both diffusion mechanisms (intra-particle and film) could be involved in the biosorption process and could be the determining rate stage of the process. The fact that the graphic representation consists of two right segments indicates that two or more stages controlled the biosorption process [37,43]: (a) the first part could be regularly associated with film diffusion (external mass transfer) [39]; (b) the second linear part suggests there is intra-particles diffusion (in the porous structure of the adsorbent) [44].The graphical representation of the McKay model (Figure 7e) shows that the diffusion of the film is not involved in the biosorption process because the graph *ln* (1-*F*) vs. *t* is not linear, and the fact that the line does not pass through the origin indicates that it is not the step that limits the (bio)sorption process rate. Consequently, the rate-limiting step would be in the first minutes the mass transfer of Co^2+^ ions at the external surface of OxC25 hydrogel associated after with film diffusion and followed by intraparticle diffusion.

Moreover, the behavior of this hybrid hydrogel based on PAV and cellulose for retaining of Co^2+^ ions from aqueous media (*q_e,max_* = 181.82 mg/g), is comparable with that of other types of hydrogels used as biosorbent in removal processes of certain metal ions from aqueous solutions, such as: (i) PVA-P (AA-co-AM) semi-IPN hydrogel—Co^2+^; q = 1563.13 mg/g [35]); (ii) chitosan-CMC—Cu^2+^; q = 169.49 mg/g [45]; (iii) interpenetrating network hydrogel—Ni^2+^; q = 102.34 mg/g [46]; (iv) starch—g-poly(acrylicacid) (PAA)/sodium humate (St-g-PAA/SH) hydrogels—Cu^2+^; q = 177.9 mg/g [47]; (v) polycarboxilated starch-based hydrogel—Cu^2+^; q = 128.26 mg/g [48]; (vi) N-vinyl imidazole-based hydrogel—Pb^2+^; q = 30.38 mg/g [49]; (vii) hydrogel based on chitosan-glucose andacrylic acid—Cu^2+^; q = 286 mg/g and Co^2+^; q = 273.9 mg/g [50]; (viii) tailored chitosan/orange peel hydrogel composite—Cu^2+^; q = 116.637 mg/g and hydrogel—Cr^6+^; q = 107.498 mg/g [51].

Taking into account all these aspects, it may be possible to consider that (bio)sorption using the OxC25 hydrogel can be a viable alternative solution for retaining of Co(II) ions from an aqueous system under specific conditions.

## 4. Conclusions

The obtained results reinforce the fact that hydrogels based on cellulose crosslinked poly (vinyl alcohol) can be considered as materials with effective biosorbent properties in retaining metal ions present in aqueous solutions in moderate concentrations. The process of Co^2+^ ions (bio)sorption was studied from the point of view of the operational factors that can influence it (biosorbent concentration, pH of the solution, temperature and contact time of the phases). Thus, the (bio)sorption process proceeds satisfactorily in an acidic environment (pH = 1.5), with a biosorbent concentration of 2 g/L, initial concentrations of moderate cations, chosen according to the data on their presence in wastewater, at temperatures of 20–35 °C and a time contact of 500 min.

Kinetic modeling was done considering some of the most used kinetic models presented in the scientific literature: Lagergreen, Ho, Elovich, Webber–Morris and KcKay. The results highlighted the complexity of the (bio)sorption mechanism on the OxC25 type hydrogel, a process controlled by the diffusion processes that can take place using either intra-particle diffusion or a film diffusion model. (Bio)sorption kinetics of Co^2+^ ions onto OxC25 hydrogels follows a pseudo-second order kinetic model, and data processing according to Webber and McKay models showed that the film diffusion is not significantly involved in the biosorption process, and intra-particles diffusion is predominant.

## Figures and Tables

**Figure 1 polymers-13-01444-f001:**
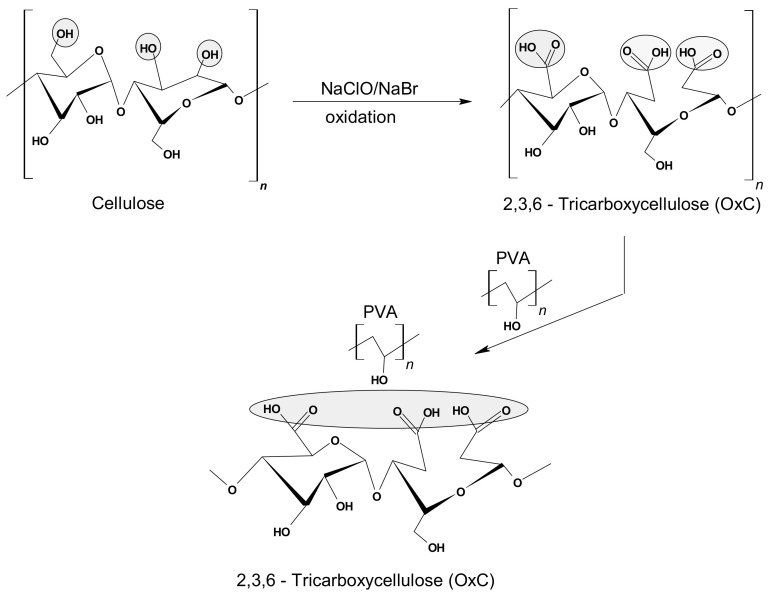
General synthesis scheme of the hybrid hydrogel based on PVA and 2,3,6-tricarboxycellulose.

**Figure 2 polymers-13-01444-f002:**
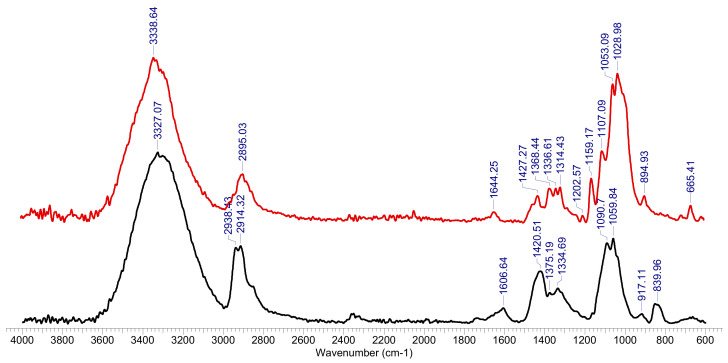
FTIR spectrum for OxC25 hydrogel before (line black) and after Co^2+^ species adsorption (red line).

**Figure 3 polymers-13-01444-f003:**
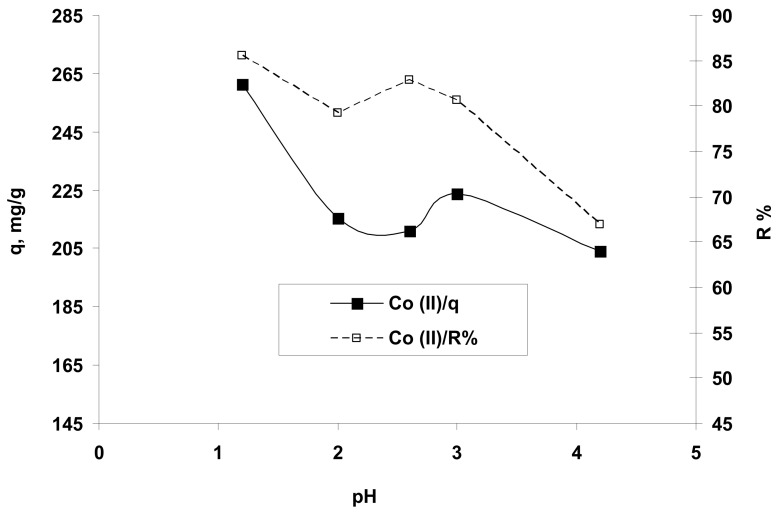
The influence of the solution pH on the retention of Co^2+^ ions on hydrogel OxC25. Operational conditions: *C*_0_ = 122.07 mg/L; 0.4 g/L biosorbent; T = 20 °C.

**Figure 4 polymers-13-01444-f004:**
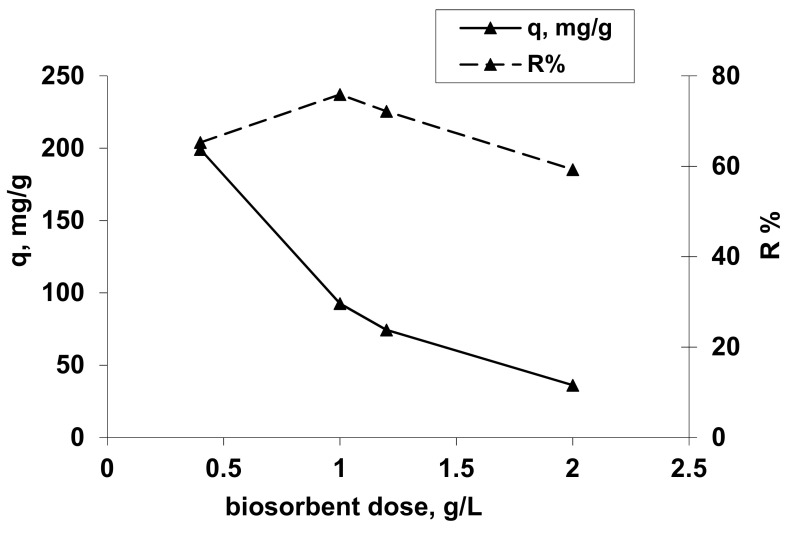
Influence of biosorbent concentration on Co^2+^ uptake. Operational conditions: *C*_0_ = 101.725 mg/L; pH =1.5; contact time = 24 h, T = 20 °C.

**Figure 5 polymers-13-01444-f005:**
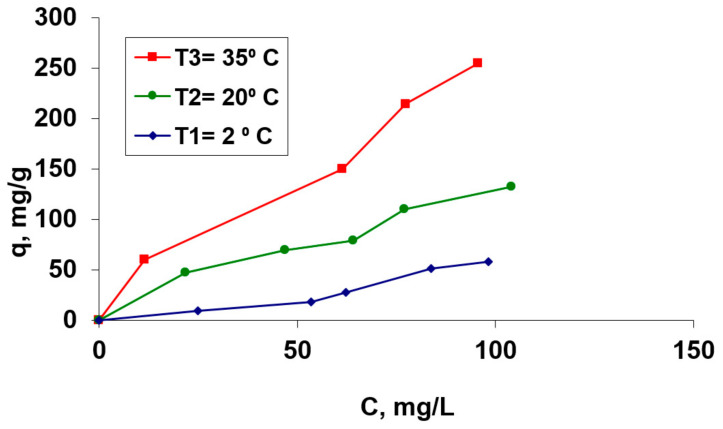
Isotherms of Co(II) biosorption process on the OxC25 hydrogel as a biosorbent. Operational conditions: *C*_0_ = 36.62 mg/L to 207.52 mg/L; pH =1.5; contact time = 24 h, biosorbent concentration = 2 g/L.

**Figure 6 polymers-13-01444-f006:**
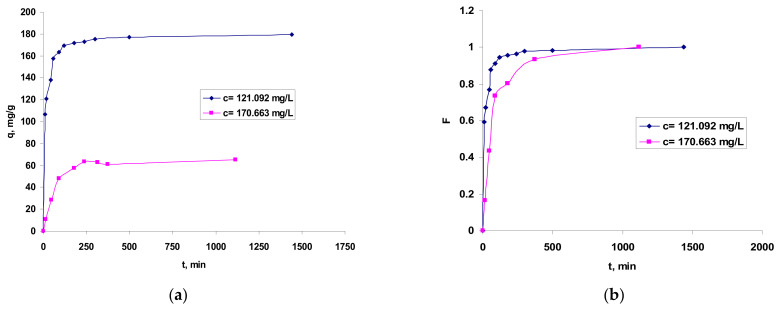
Influence of contact time in Co^2+^ ions biosorption onto OxC25 hydrogel vs. (bio)adsorption capacity (**a**) and fractional equilibrium attainment (**b**). Operational conditions: adsorbent concentration of 0.4 g/L; pH = 1.5; temperature of 20 °C.

**Figure 7 polymers-13-01444-f007:**
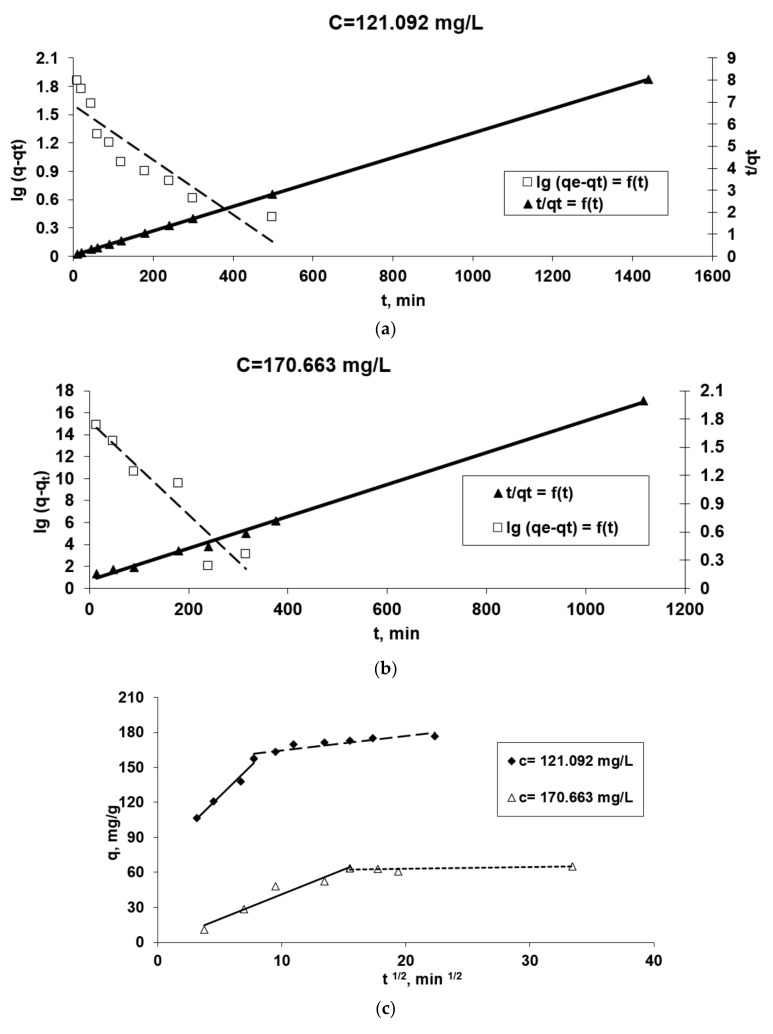
The applicability of some kinetic models [Pseudo-first order and pseudo-second order diffusion kinetic (**a**,**b**), Webber (**c**), Elovich (**d**) and McKay (**e**)] in the Co^2+^(bio)sorption onto OxC25 hydrogel. Operational conditions: biosorbent concentration of 0.4 g/L; pH = 1.5; temperature of 20 °C; two initial concentrations of Co(II) solution (C_Co2+_ = 121.092 mg/L, or 170.663 mg/L).

**Table 1 polymers-13-01444-t001:** Kinetic parameters for the Co^2+^ ions biosorption onto OxC25 hydrogel.

Kinetic Model/Characteristic Parameters	Initial Co^2+^ Concentration, mg/L	
	121.902	170.663
**Pseudo First Order Kinetic**		
*k*_1_, (1/min)	3.684	4.096
*q_e_*, (mg/g) *—* Biosorption capacity cannot be calculated using the experimental data from graphical representations because, according to the value of R^2^, the data do not enable use of this kinetic model.		
R^2^	0.8234	0.8929
**Pseudo Second Order Kinetic**		
*k*_2_, [g/(mg min)]	0.000293	0.01996
*q_e_*, (mg/g)	181.82	68.965
R^2^	1.000	0.9971
**ElovichModel**		
α, [mg/(g min)]	3.502	4.362
β, (g/mg)	0.0516	0.07302
R^2^	0.9143	0.8904
**Webber—Intra-Particle Diffusion**		
k_d1_, [mg/(g min^0.5^)]	10.402	4.2613
R^2^	0.9732	0.9392
*k_d_*_2_, [mg/(g min^0.5^)]	1.2394	0.1438
R^2^	0.803	0.4318
**McKay**		
R^2^	0.8955	0.8639

*q_e,exp_* (mg/g) 181.82 *; * data being published.

## Data Availability

All data for experimental results are presented in this study. No archived datasets analyzed or generated during the study for public consulting.

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
