# Peer review of "Hydrogel Based on Tricarboxi-Cellulose and Poly(Vinyl Alcohol) Used as Biosorbent for Cobalt Ions Retention"

_polymers, 2021, doi:10.3390/polym13091444_

Round 1

Reviewer 1 Report

The manuscript from the title: "Hydrogels based on tricarboxi-cellulose and poly (vinyl alcohol) used as biosorbent for cobalt ions retention", is a very interesting paper for read. However, it is necessary to add and corrected the following data:

1. In Figure 2, it is said that the diagram is qads = f (pHi), and the diagram shows the amount of adsorbent on the x-axis, the figure should be replaced with the appropriate one. Replace Figure 2 and Figure 4 with each other.

Did the authors examine the stability of the hydrogel at pH=1-2? If so, these results should be shown.

2. Lines 303-313 - Did the authors monitor the pH change during the adsorption process? Given that at very low pH values the concentration of H+ or H3O+ is very high, can the authors give an explanation of whether there was influence of H+ or H3O+ ions to remove cobalt from aqueous solution under these experimental conditions? How can the increase in removal efficiency obtained at pH=3 be explained?

3. In the title of chapter 3.2.4. it was stated that the influence of dye concentration and temperature was studied, but the results of the influence of dye on the efficiency of cobalt removal were not shown. Please explain that.

4. In addition, no thermodynamic parameters have been determined, nor have appropriate explanations been given. Please explain that.

5. The conclusions given in the kinetic tests are not related to the test results of the influence of pH and mass of the adsorbent, and as such the results do not act as a whole.

6. For Figure 6e, the parameters in Table 3 are not shown.

I recommend accept of paper after minor revision (corrections to minor methodological errors and text editing).

Author Response

Dear Reviewer #1,

Thank you for your time spent for reviewing our manuscript and for your comments and suggestions that have been helpful to improve our work. We revised the manuscript and all changes were highlighted in red.

The list of responses to your comments is attached.

Reviewer 2 Report

The presented study reports on the hydrogels is used as biosorbent for cobalt ions retention, and a proper series of experimental results have been carried out to support the claim of this work. After carefully reading it, this study is a useful and meaningful work, and please consider the following points to try to improve the quality of it.

1 There is no subject for this sentence in Introduction of “Thus were developed new treatment concepts of reuse/recycling…”

2 The introduction is so long, it is suggested to summarize it.

3 “3.2.1. Evaluation … parameters’ influence” is revised to “3.2.1. Evaluation … parameters’ influences”

4. Figure 1 should be improved, the details for x and y coordinates should be prepared.

5. The working principles of the biosorption is not enough. As the Co(II) ions diffusion into the hydrogel, I suggest to consider the following references, (1) Hiroyuki Kamata, Yuki Akagi, Yuko Kayasuga-Kariya, Ung-il Chung, Takamasa Sakai. “Nonswellable” Hydrogel Without Mechanical Hysteresis. SCIENCE. 2014, 343, 873-875. (2) Haibao Lu, Yanju Liu, Jinsong Leng and Shanyi Du. Qualitative Separation of the Physical Swelling Effect on the Recovery Behavior of Shape Memory Polymer. European Polymer Journal. 2010, 46(9): 1908-1914. (3) Haibao Lu and Shanyi Du. A phenomenological thermodynamic model for the chemo-responsive shape memory effect in polymers based on Flory-Huggins solution theory. Polymer Chemistry. 2014, 5(4), 1155-1162.

It should be clearly pointed out that the hydrogel is swollen or nonswelling due to diffusion of Co(II) ions, with the water molecules? Or the diffusion is governed by the pseudo-second order kinetic mode, it is governed by the Flory-Huggins solution theory of hydrogel, whereas the Co(II) ions is absorbed? In the current version, it is difficult to find out the working principles behind these experimental results.

6. Discussion on the kinetic model is not enough. Based on figure 6, it is difficult to catch up with that the Pseudo second order kinetic model is better. Please compare the analytical results of the model in a figure for clearly read.

Author Response

Dear Reviewer #2,

Thank you for your time spent for reviewing our manuscript and for your comments and suggestions that have been helpful to improve our work. We revised the manuscript and all changes were highlighted in red.

The list of responses to your comments is attached.

Reviewer 3 Report

This article reported the synthesis of hydrogels and their applications for the adsorption of Co(II) ions from water. The topic is interesting. However, the manuscript needs significant revision before considering it for publication. It is not recommended for publication in the current form. The authors need to carefully revise and improve the quality of the manuscript before re-submission.

Below are my comments:

  1. The abstract can be improved and some key results must be highlighted with numerical values such as max adsorption capacity.
  2. The introduction is very lengthy and a bit scattered. Some paragraphs might be shortened or deleted such as paragraphs 2 and 3. The introduction should briefly highlight the water pollution, associated health and environmental problems with heavy metals especially Co(II), traditional techniques for the removal of metals (or Co(II) specifically), advantages of adsorption, some commonly sued adsorbents and then establish the novelty/advantages of using the proposed hydrogels. Likewise, some re-arrangement is needed. Paraphs 6-8 can be merged and should be placed at the start (second/third paragraph). I will suggest reducing the length (around one-third of the current) to make it more clear and concise.
  3. Section 2.1, needs to be rephrased. The first sentence is not clear.
  4. Section 2.2, it will be good if a schematic of the synthesis of adsorbent is provided.
  5. Use the full form of abbreviations when using for the first time such as SEM, FTIR, TEMPO
  6. Equation 1, provide a single equation for the adsorption capacity.
  7. Keep a uniform terminology, such as Co(II) or Cobalt(II) or Co2+ (See line 248, 252 and 268)
  8. Section 3.2.2 and 3.2.3, figures 2 and 4 are wrongly quoted. Figure 2 is the effect of dosage.
  9. What was the point of zero charge of the adsorbent? Without knowing it, the effect of pH and adsorption mechanism cannot be explained accurately.
  10. Like FTIR image it is better to provide SEM images of all materials.
  11. The figures should be provided in the same format with the same font size. Also, pay attention to subscripts and superscripts.
  12. Why isotherm study was not conducted to get an idea of the predicted adsorption capacity?
  13. What was BET surface area of the adsorbent?
  14. It is recommended to provide a table to compare the performance of the prepared adsorbent with the material reported in the literature for the adsorption of Co(II).
  15. The conclusion should be different from the abstract. It should be concise and only the key findings should be highlighted.
  16. The following relevant papers are suggested to be included.
  • https://doi.org/10.1016/j.jiec.2016.07.012
  • https://doi.org/10.1007/s13738-019-01738-8
  • 1039/C9RA00227H
  • https://doi.org/10.1016/j.seppur.2015.11.039
  • https://doi.org/10.1016/j.chemosphere.2020.129415

Author Response

Dear Reviewer #3,

Thank you for your time spent for reviewing our manuscript and for your comments and suggestions that have been helpful to improve our work. We revised the manuscript and all changes were highlighted in red.

The list of responses to your comments is attached.

Reviewer 4 Report

The manuscript reports on the use of a hydrogel based on tricarboxi-cellulose and poly (vinyl alco-2 hol) as biosorbent for cobalt ions. The adsorption properties of the biosorbent have been evaluated in batch experiments in presence of Co2+ ions. In my opinion the work plan is correct, but some improvements are needed. The manuscript could benefit by a linguistic revision since some sentences are not clear and too long and have to be rewritten. Figures and Table should be improved. The introduction is too long and the experimental section should be re-organized and improved. The manuscript in not suitable for publication to me in the present form and major revisions are recommended. In the following a list of comments and issues to be addressed in the revised version of the manuscript.

Entire manuscript:

  • The entire manuscript needs a linguistic revision

Title

  • Avoid the use of plural since only one material is the object of the work

Introduction

  • It is too long. Obviously in the present form it contains a lot of important information, but it is too dispersive. I suggest to reduce it focusing on the use of hydrogels for heavy metal removals from wastewaters. I also suggest to underline the concentrations of the target pollutant in the wastewaters and the values admitted by the international rules.

Experimental section

  • Add a reference for the cellulose oxidation (lines 219-222)
  • Is the range of Co2+ concentration explored in line with the values of [Co2+] typical of wastewaters?
  • The preparation of the hydrogel is object of previous publications (refs 28 and 29), anyway I suggest to report a brief description of the synthetic procedure, also because some difficulties in the accessibility to the previous publications can be encountered.
  • Line 245: please report the correct ion
  • Did the authors perform repeated tests on the material to establish its reusability? How the material can be regenerated?
  • Have the authors proofs of the stability in water of the material? I mean, are the authors sure that the material does not decompose after a period of usage? Swelling phenomena can take place? Did the authors check the structural properties of the hydrogel after its usage?
  • Which is the reproducibility of the synthetic procedure and of the adsorption tests?
  • The adsorption experiments have been performed in a static fashion or such a kind of shaking has been applied (orbital shaking for example)?
  • Why only two Co2+ concentrations have been explored? Why a complete isotherm exploring more Co2+ concentrations has not been measured? Which is the qmax of the biosorbent?

Results and discussion

  • Table 1: the data reported in it are already reported in a previous publication of the same authors. I suggest to remove the table, to report the figures as supporting information and to add a summary of the previous findings clearly stating that the characterization of the material was object of previous works.
  • Lines 257-266: move to the experimental section
  • Figure 1: improve it, labelling the most important bands to highlight better the changes occurring after the biosorption.
  • Description of Figure 1 (lines 269-279): improving it also explaining better the effects related to the interaction with Co2+ ions. Some differences arise also for the bands at 1500 cm-1. Is the spectrum of neat biosorbent taken after contact with pure water? How the spectrum of the hydrogel changes after the contact with water?
  • Lines 282-289: please check and rewrite
  • Line 300: what does it mean “redox conditions of aqueous environment”?
  • Figure 2 and Figure 4: please invert the two graphics
  • Lines 308-313: please check. At low pH the carboxylic groups are expected to be in the COOH form or not?
  • Figure 3: please improve it. In the present form and in that position is not informative. I suggest to move it in the experimental section and I also suggest to modify it, highlighting not only the structure of the two components but also the linkages between them in the final product.
  • Lines 338-340: specify that the data in table 2 refers also to different temperatures
  • Table 2 and lines 344-351: is there a possible explanation for the change from an exothermic to an endothermic process?
  • Lines 353-357: please check and rewrite
  • Why only two Co2+ concentrations have been tested and not a higher number allowing for the construction of an isotherm and for the determination of qmax?
  • Line 363: what is the "fractional attainment of equilibrium (F)” and how it is calculated?
  • Section 3.3: the theoretical details of the different models used for analyzing the adsorption data should be moved to the experimental section. A subsection about modeling theory can be there introduced.
  • Final part of Section 3.3: since there is a Conclusions section I suggest to avoid the use of terms related to conclusions, namely I suggest to change “underline some conclusions” at line 427 and “Concludingly” at line 464.

Author Response

Dear Reviewer #4,

Thank you for your time spent for reviewing our manuscript and for your comments and suggestions that have been helpful to improve our work. We revised the manuscript and all changes were highlighted in red.

The list of responses to your comments is attached.

Round 2

Reviewer 3 Report

This is the revised version of the manuscript. Unfortunately, I have not seen a major improvement in the revised version. Many comments are not addressed properly. I cannot recommend the manuscript unless it is properly revised by the authors.

Below are the comments that need to be addressed:

  1. The abstract can be improved and some key results must be highlighted with numerical values such as max adsorption capacity (not revised accordingly).
  2. The introduction is very lengthy and a bit scattered. Some paragraphs might be shortened or deleted such as paragraphs 2 and 3. The introduction should briefly highlight the water pollution, associated health and environmental problems with heavy metals especially Co(II), traditional techniques for the removal of metals (or Co(II) specifically), advantages of adsorption, some commonly sued adsorbents and then establish the novelty/advantages of using the proposed hydrogels. Likewise, some re-arrangement is needed. Paraphs 6-8 can be merged and should be placed at the start (second/third paragraph). I will suggest reducing the length (around one-third of the current) to make it clearer and more concise (Still lengthy, needs improvement and should be concise).
  3. Section 2.1, needs to be rephrased. The first sentence is not clear (the same issue exists).
  4. Section 2.2, it will be good if a schematic of the synthesis of adsorbent is provided (Schematic is good, however, provide a high-quality image).
  5. Use the full form of abbreviations when using for the first time such as SEM, FTIR, TEMPO (same issue)
  6. What was the point of zero charge of the adsorbent? Without knowing it, the effect of pH and adsorption mechanism cannot be explained accurately (Discuss the effect of pH with reference to point of zero charge, surface chemistry of the adsorbent and cobalt species in aqueous medium under various pH).
  7. It is recommended to provide a table to compare the performance of the prepared adsorbent with the material reported in the literature for the adsorption of Co(II) (Provide a table to compare with other adsorbents not necessary with hydrogels).

Author Response

Dear Reviewer #3,

We revised our 1st revised manuscript considering all your comments and recommendations. 

Reviewer 4 Report

The authors addressed the comments and the questions I formulated during the first round of revision improving the manuscript accordingly. The revised version of the manuscript is more complete and readable. The manuscript is suitable for publication to me in the present form.

Author Response

Dear Reviewer #4,

We thank you for your recommendations and suggestions helping us to improve the quality of our manuscript. We are happy that our 1st revised manuscript format was found suitable for publication in this journal.

Good wishes.

Round 3

Reviewer 3 Report

The revised manuscript can be considered for publication. Still, I believe that some issues can be addressed at the proofing stage.

  1.  The abstract is still very general. The authors can improve it by including more specific details about this study.
  2. Section 2.1 can be rephrased it. “A hybrid hydrogel with two basic components: poly(vinyl alcohol) (PVA) and 2,3,6 tricarboxycellulose, prepared by oxidation of the wet cellulose  pulp in the presence of the 2,2,6,6-tetramethyl-1-piperidinyloxy (TEMPO) / NaBr / NaOCl system, was used as a biosorbent”.
  3. The equations are not visible clearly.
  4. Section 3.2.2, the authors need to make sure if Co2+ is the dominant species present in water at different pH.